# Privacy-Preserving Adaptive Tracking Control for Multiagent Systems

1st Siyu Guo
*College of Control Science and Engineering*
*Bohai University*
Jinzhou, China
2022008033@qymail.bhu.edu.cn

*Abstract*—**This paper investigates the privacy-preserving adaptive tracking control problem for nonlinear multiagent systems with unknown disturbances. Firstly, based on the phenomenon of Lorenz system, a masking function endowed with uncertainty and unpredictability is proposed. Meanwhile, a time-assist function is designed to achieve privacy protection within a predefined period of time, which enhances the flexibility of the mechanism. In addition, a dynamically self-adjustable gain parameter is designed in disturbance observer. The introduction of this parameter enables the proposed disturbance observer to not only handle unknown disturbances but also achieve better results in improving system performance. Afterward, the Lyapunov stability theorem is utilized to prove that all signals of the closed-loop system are semi-globally uniformly ultimately bounded. Finally, simulation results confirm the effectiveness of the proposed control scheme.**

*Index Terms*—**Disturbance observer, gradient descent method, multiagent systems, privacy-preserving mechanism.**

## I. INTRODUCTION

In recent years, multiagent systems (MASs) have garnered ongoing attention across diverse fields [1, 2]. In the control problems of MASs, effective communication between agents is crucial to accomplish common control tasks. Most communication channels frequently lack protection, leaving communication networks susceptible to malicious attacks. Additionally, each agent has sensitive information that it prefers to keep private. Hence, there arises an urgent necessity for integrating privacy protection attributes into MASs.

Given the hidden risks in information interaction, some scholars have increasingly directed their attention to researching privacy protection in MASs [3, 4]. At present, the mainstream method is to design a protection function with masking ability. Notably, the parameters in these masking functions are fixed, which increases the risk of the specific form of the masking function being deciphered. Furthermore, most of the current masking functions are applied in infinite time domains, which may not only affect the stability of the system but also lack the ability to flexibly choose protection time. The question of enhancing the protective effectiveness of masking functions and achieving privacy protection within user-defined time frames is worth exploring, and it has inspired the research presented in this paper.

Unknown disturbances are prevalent in industrial applications, posing a threat to the stability of the system. To effectively detect and mitigate the impact of these disturbances, disturbance observer has been extensively researched [5, 6]. Most of the design methods involved in the disturbance observers lack certain adaptability. In order to improve the performance of the disturbance observers, some scholars have delved into the correlation between the disturbance observers and system response [7]. This prompts us to further investigate the interplay between the disturbance observers and system errors, with the goal of detecting unknown disturbances while optimizing system performance.

Motivated by the above discussion, this paper proposes a preassigned-time hyperchaotic privacy-preserving mechanism and an optimization-based adaptive disturbance observer. The contributions are summarized as follows.

1) Based on the characteristics of Lorenz system, the designed masking function is endowed with unpredictable stochastic properties, which enhances the masking effectiveness and protective functionality. Furthermore, by designing a time-assist function, privacy protection within predefined time-frames is achieved, which mitigates the adverse impact of protection-induced uncertainty on system performance.

2) An adaptive gain parameter is designed in the disturbance observer based on the gradient descent method. The introduced adaptive gain parameter enables the disturbance observer not only to detect and estimate unknown disturbances but also to achieve the effect of improving the control performance of the MASs.

## II. PRELIMINARIES

### A. Graph Theory

Consider a directed graph $\mathsf{G} = (\mathsf{V}, \mathsf{E}, \mathsf{A})$ with $\mathsf{M}$ nodes. $\mathsf{V} = (1, 2, \ldots, \mathsf{M})$ is a non-empty set of agents. $\mathsf{E} \in \mathsf{V} \times \mathsf{V}$ is an edge set, where $(\mathsf{V}_h, \mathsf{V}_j) \in \mathsf{E}$ is an edge from node $j$ to node $h$. $\mathsf{A} = [p_{h,j}] \in \mathbb{R}^{\mathsf{M} \times \mathsf{M}}$ represents the adjacency matrix. If $(\mathsf{V}_h, \mathsf{V}_j) \in \mathsf{E}$, $p_{h,j} = 1$, otherwise $p_{h,j} = 0$. The Laplacian matrix is defined as $\mathsf{L} = \mathsf{D} - \mathsf{A}$, including the in-degree matrices $\mathsf{D} = \mathrm{diag}(d_1, \ldots, d_\mathsf{M})$ with $d_h = \sum_{j=1}^{\mathsf{M}} p_{h,j}$. Define $\mathsf{B} = \mathrm{diag}(b_1, \ldots, b_\mathsf{M})$. The positive and negative of $b_h$ represent whether the node $h$ is able to obtain information directly from the leader node or not, respectively.

*Lemma 1:* [1] There exists a spanning tree in the graph G, where the leader node 0 is designated as the root node. With this, L + B is nonsingular.

### B. Problem Formulation

In this paper, we examine nonlinear MASs comprising M followers and a leader, where the dynamic of the $h$th ($h = 1, \ldots, M$) follower is characterized by

$$
\begin{cases}
\dot{x}_{h,m} = x_{h,m+1} + j_{h,m}(\bar{x}_{h,m}) \\
\dot{x}_{h,n} = u_h + j_{h,n}(\bar{x}_{h,n}) + \omega_h(t) \\
y_h = x_{h,1}
\end{cases} \tag{1}
$$

where $m$ represents the order of the system and satisfies $1 \leq m < n-1$, $\bar{x}_{h,m} = [x_{h,1}, x_{h,2}, \ldots, x_{h,m}]^{\mathrm{T}} \in \mathbb{R}^m$ and $\bar{x}_{h,n} = [x_{h,1}, x_{h,2}, \ldots, x_{h,n}]^{\mathrm{T}} \in \mathbb{R}^n$ indicate the state vectors of the $h$th follower. $u_h$ is the control input signal, and the output signal of the $h$th follower is expressed as $y_h$. $\omega_h(t)$ is the unknown disturbance. $j_{h,m}(\bar{x}_{h,m})$ and $j_{h,n}(\bar{x}_{h,n})$ are unknown nonlinear functions. $y_r(t)$ stands for the leader signal of the MASs. Then, the synchronization error of the $h$th follower is given by $z_{h,1} = \sum_{j=1}^{M} p_{h,j}(y_h - y_j) + b_h(y_h - y_r)$.

*Lemma 2:* [7] Define $\acute{z}_{h,1} = (z_{1,1}, \ldots, z_{M,1})^{\mathrm{T}}$, $\acute{y}_h = (y_1, \ldots, y_M)^{\mathrm{T}}$, $\acute{y}_r = (y_r, \ldots, y_r)^{\mathrm{T}}$. Then, it follows that $\|\acute{y}_h - \acute{y}_r\| \leq \frac{\|\acute{z}_{h,1}\|}{\bar{\mathcal{M}}(L+B)}$, where $\bar{\mathcal{M}}(L+B)$ denotes the minimum singular value of L + B.

*Lemma 3:* [8] The second-order sliding mode integral filter (SSMIF) is established as

$$
\begin{cases}
\dot{\mathbf{K}}_{h,k10} = -\dfrac{\mathbf{K}_{h,k10} - \mathbf{w}(t)}{\mathbf{k}_{h,k10}} - \dfrac{\mathbf{R}_{h,k10}(\mathbf{K}_{h,k10} - \mathbf{w}(t))}{\|\mathbf{K}_{h,k10} - \mathbf{w}(t)\| + \iota_{h,k10}} \\
\dot{\mathbf{K}}_{h,k20} = -\dfrac{\mathbf{K}_{h,k20} - \mathbf{K}_{h,k10}}{\mathbf{k}_{h,k20}} - \dfrac{\mathbf{R}_{h,k20}(\mathbf{K}_{h,k20} - \mathbf{K}_{h,k10})}{\|\mathbf{K}_{h,k20} - \mathbf{K}_{h,k10}\| + \iota_{h,k20}}
\end{cases} \tag{2}
$$

where $\mathbf{K}_{h,k10}$ and $\mathbf{K}_{h,k20}$ denote the states of the filter (2). $\mathbf{k}_{h,k10}$, $\mathbf{k}_{h,k20}$, $\iota_{h,k10}$ and $\iota_{k,k20}$ are positive design parameters.

*Lemma 4:* [5] One can approximate an unknown function $P(s)$ by using Radial Basis Function Neural Networks (RBF NNs). $P(s) = \zeta^{*\mathrm{T}}\phi(s) + \varepsilon(s)$, where $\phi(s) = [\phi_1(s), \phi_2(s), \ldots, \phi_r(s)]^{\mathrm{T}}$ denotes the basis function vector. $\varepsilon(s)$ represents the approximation error. The ideal weight vector $\zeta^* = [\zeta_1^*, \zeta_2^*, \ldots, \zeta_r^*]^{\mathrm{T}} \in \mathbb{R}^r$ can be chosen as $\zeta^* = \arg\min_{\zeta \in \mathbb{R}^r} \{\sup_{s \in \Omega} |P(s) - \zeta^{\mathrm{T}}\phi(s)|\}$.

## III. MAIN RESULTS

### A. The Preassigned-Time Hyperchaotic Protection Mechanism

In this paper, a class of Lorenz-Stenflo hyperchaotic systems are considered, which is modelled as

$$
\begin{cases}
\dot{\underline{x}} = a(\underline{y} - \underline{x}) + \mu\underline{w} \\
\dot{\underline{y}} = c\underline{x} - \underline{y} - \underline{x}\underline{z} \\
\dot{\underline{z}} = -\beta\underline{z} + \underline{x}\underline{y} \\
\dot{\underline{w}} = -\underline{x} - a\underline{w}
\end{cases} \tag{3}
$$

where $a$ is the Prandtl number, $c$ is the generalized Rayleigh parameter, $\beta = 4k_1^2/k_2^2$ and $\mu = 4\Omega^2 k_1^2/k_h^2 k_2^6$, $\Omega$ is the angular frequency of the earth rotation, $k_h$ is the heat dissipation

coefficient, $k_1$ and $k_2$ are design constants. In this paper, we choose $a = 3$, $\mu = 30$, $c = 30$ and $\beta = 8/3$.

Next, a time-assist function $\Pi(t)$ is designed to achieve privacy protection within a specified timeframe. The specific expression of the function is as follows:

$$
\Pi(t) = \begin{cases}
1 + \left[\cos\left(\frac{\pi t}{T_\epsilon}\right)\right]^{(2n+1)} & 0 \leq t < T_\epsilon \\
0 & t \geq T_\epsilon
\end{cases} \tag{4}
$$

where $T_\epsilon$ is the user-defined protection time with arbitrary settings, and $2n + 1$ is the designed function order. Correspondingly, it is concluded that

$$
\dot{\Pi}(t) = \begin{cases}
-\frac{(2n+1)\pi}{T_\epsilon}\left[\sin\left(\frac{\pi t}{T_\epsilon}\right)\right]^{2n} & 0 \leq t < T_\epsilon \\
0 & t \geq T_\epsilon
\end{cases} \tag{5}
$$

Based on the above analysis, a masking function is designed for the leader signal, which is expressed as

$$
y_\pi = y_r + \left[\kappa\left(\int(-\underline{x} - a\underline{w})\mathrm{d}t\right)e^{-\varsigma(\int(a\underline{y} - a\underline{x} + s\underline{w})\mathrm{d}t)t}\right]\Pi(t) \tag{6}
$$

where $\kappa$ and $\varsigma > 0$ are designed parameters.

### B. The Optimization-Based Adaptive Disturbance Observer

The optimization-based adaptive disturbance observer is designed in the following form [5]:

$$
\begin{cases}
\dot{\hat{\bar{\omega}}}_h = \kappa_h(x_{h,n} - \xi_h) \\
\dot{\xi}_h = u_h + \ell_h\hat{\bar{\omega}}_h + \hat{\zeta}_{h,n}^{\mathrm{T}}\phi_{h,n}(\bar{x}_{h,n})
\end{cases} \tag{7}
$$

where $\hat{\zeta}_{h,n}$ and $\phi_{h,n}(\bar{x}_{h,n})$ denote the approximate parameter vector and basis function vector, respectively. $\kappa_h$ means a positive design parameter. $\hat{\bar{\omega}}_h$ is the estimated value of $\bar{\omega}_h$ with $\tilde{\omega}_h = \bar{\omega}_h - \hat{\bar{\omega}}_h$, and the specific form of $\bar{\omega}_h$ will be given later. $\ell_h$ is an iterative parameter that can be varied adaptively. After that, it can be deduced that

$$
\dot{\tilde{\omega}}_h = \dot{\bar{\omega}}_h - \kappa_h\left(\ell_h\tilde{\omega}_h + \tilde{\zeta}_{h,n}^{\mathrm{T}}\phi_{h,n}(\bar{x}_{h,n})\right) \tag{8}
$$

with $\dot{\bar{\omega}}_h$ being a bounded value. The controller $u_h$ is designed in the form of $u_h = -(\vartheta_{h,n} + \frac{1}{2})\delta_{h,n} + \mathbf{K}_{h,n20} - \delta_{h,n-1} - \ell_h\hat{\bar{\omega}}_h - \hat{\zeta}_{h,n}\phi_{h,n}(\bar{x}_{h,n})$. The detailed design process of $u_h$ will be analyzed later. Then, according to the error transformation relation $\delta_{h,n} = x_{h,n} - \alpha_{h,n-1}$, the derivative of the error variable $\dot{\delta}_{h,n}$ satisfies $\dot{\delta}_{h,n} = u_h + f_{h,n} + \omega_h(t) - \dot{\alpha}_{h,n-1}$.

From this, a gradient expression for the derivative of the error variable $\dot{\delta}_{h,n}$ with respect to $\ell_h$ is obtained as $\frac{\partial(\dot{\delta}_{h,n})}{\partial(\ell_h)} = -\hat{\bar{\omega}}_h$, which reveals a potential link between $\dot{\delta}_{h,n}$ and $\ell_h$, inspiring us to improve system performance by controlling the variation of $\ell_h$.

The gradient descent method is applied to realize the self-regulation function of $\ell_h$ and improve the performance of the MASs. The specific iterative expression is given as $\ell_h = \ell_h^{\mathrm{Pr}} + s_h^{\mathrm{L}}\Xi_h(\delta_{h,n})\hat{\bar{\omega}}_h$, where $\ell_h^{\mathrm{Pr}}$ is the position of $\ell_h$ after the last iteration, and the initial position of $\ell_h$ is artificially designed. $s_h^{\mathrm{L}}$ is the step size of each iteration. $\Xi_h$ is a sign function related to $\delta_{h,n}$, when $\delta_{h,n} > 0$, $\Xi_h = 1$; when $\delta_{h,n} < 0$, $\Xi_h = -1$.

## IV. ADAPTIVE CONSENSUS CONTROL OF MASS

### A. Adaptive Controller Design

In order to accomplish the proposed control objective, the following common error transformation is implemented, specified as

$$\begin{cases} \delta_{h,1} = \sum_{j=1}^{\mathsf{M}} p_{h,j}(y_h - y_j) + b_h(y_h - y_\pi) \\ \delta_{h,g} = x_{h,g} - \alpha_{h,g-1} \quad g = 2,3,\ldots,n \end{cases} \quad (9)$$

where $\alpha_{h,g-1}$ is the virtual control signal.

**Step 1.** The Lyapunov function is selected as $V_{h,1} = \frac{1}{2}\delta_{h,1}^2 + \frac{1}{2\varrho_{h,1}}\tilde{\zeta}_{h,1}^2$, where $\varrho_{h,1} > 0$ is a design constant. In view of the unknown function $F(\acute{X}_h) = \bar{p}_h\jmath_{h,1} - \sum_{j=1}^{\mathsf{M}} p_{h,j}(x_{j,2} + \jmath_{j,1})$ with $\acute{X}_h = [x_{h,1}, x_{j,1}, x_{j,2}]^{\mathrm{T}}$, the RBF NN is introduced to approximate $F(\acute{X}_h)$.

To achieve system stability, the following design is implemented for both the virtual control signal and the adaptive law.

$$\begin{cases} \alpha_{h,1} = -\left(\frac{2\vartheta_{h,1}+1}{2\bar{p}_h}\right)\delta_{h,1} - \frac{\hat{\zeta}_{h,1}\phi_{h,1}(\acute{X}_{h,1})}{\bar{p}_h} + \frac{b_h}{\bar{p}_h}\dot{y}_\pi \\ \dot{\hat{\zeta}}_{h,1} = \varrho_{h,1}\delta_{h,1}\phi_{h,1}(\acute{X}_{h,1}) - \chi_{h,1}\hat{\zeta}_{h,1} \end{cases} \quad (10)$$

where $\vartheta_{h,1} > 0$ and $\chi_{h,1} > 0$ are design constants.

**Step s.** For step $s$ with $s = 2,\ldots,n-1$, an SSMIF is used to estimate $\dot{\alpha}_{h,s-1}$. The Lyapunov function is selected as $V_{h,s} = V_{h,s-1} + \frac{1}{2}\delta_{h,s}^2 + \frac{1}{2\varrho_{h,s}}\tilde{\zeta}_{h,s}^2$, where $\varrho_{h,s} > 0$ is a design constant. Then, the virtual control signal and the adaptive law are designed as

$$\begin{cases} \alpha_{h,s} = -\left(\vartheta_{h,s}+1\right)\delta_{h,s} + \mathbf{K}_{h,s20} - \delta_{h,s-1} \\ \quad -\hat{\zeta}_{h,s}\phi_{h,s}\left(\bar{x}_{h,s}\right) \\ \dot{\hat{\zeta}}_{h,s} = \varrho_{h,s}\delta_{h,s}\phi_{h,s}\left(\bar{x}_{h,s}\right) - \chi_{h,s}\hat{\zeta}_{h,s} \end{cases} \quad (11)$$

where $\vartheta_{h,s}$ and $\chi_{h,s}$ are positive constants.

**Step n.** Similar to the previous treatment, through SSMIF, we have $\dot{\alpha}_{h,n-1} = \mathbf{K}_{h,n20} - \mathbf{K}_{h,f_{n-1}}$, where $|\mathbf{K}_{h,f_{n-1}}| < \bar{\mathbf{K}}_{h,f_{n-1}}$ with $\bar{\mathbf{K}}_{h,f_{n-1}} > 0$. Accounting for the effects of unknown disturbances, the Lyapunov function is chosen as $V_{h,n} = V_{h,n-1} + \frac{1}{2}\delta_{h,n}^2 + \frac{1}{2\varrho_{h,n}}\tilde{\zeta}_{h,n}^2 + \frac{1}{2}\tilde{\omega}_h^2$, where $\varrho_{h,n} > 0$ is a design constant.

With the aim of ensuring that all signals in the system are SGUUB, the final controller $u_h$ and the adaptive law $\dot{\hat{\zeta}}_{h,n}$ are devised as

$$\begin{cases} u_h = -\left(\vartheta_{h,n}+\frac{1}{2}\right)\delta_{h,n} + \mathbf{K}_{h,n20} - \delta_{h,n-1} - \ell_h\hat{\bar{\omega}}_h \\ \quad -\hat{\zeta}_{h,n}\phi_{h,n}\left(\bar{x}_{h,n}\right) \\ \dot{\hat{\zeta}}_{h,n} = \varrho_{h,n}\delta_{h,n}\phi_{h,n}\left(\bar{x}_{h,n}\right) - \chi_{h,n}\hat{\zeta}_{h,n} \end{cases} \quad (12)$$

where $\vartheta_{h,n}$ and $\chi_{h,n}$ are positive constants. Following the properties of the proposed optimization-based adaptive disturbance observer involved in (7) and (8), we derived $\dot{V}_{h,n} \leq -\sum_{o=1}^n \vartheta_{h,o}\delta_{h,o}^2 + \sum_{o=1}^{n-1} \frac{\bar{\varepsilon}_{h,o}^2}{2} + \sum_{o=2}^{n-1} \frac{\bar{\mathbf{K}}_{h,f_{o-1}}}{2} - \sum_{o=1}^{n-1} \frac{\chi_{h,o}\tilde{\zeta}_{h,o}^2}{2\varrho_{h,o}} - \left(\frac{\chi_{h,o}}{2\varrho_{h,o}} - \frac{|\kappa_h|}{2}\right)\tilde{\zeta}_{h,n}^2 + \sum_{o=1}^n \frac{\chi_{h,o}\zeta_{h,o}^2}{2\varrho_{h,o}} + \frac{1}{2}\dot{\omega}_h^2 - \frac{1}{2}\left(2\kappa_h\ell_h - \ell_h^2 - 1 - |\kappa_h|\phi_{h,n}^{\mathrm{T}}(\bar{x}_{h,n})\phi_{h,n}(\bar{x}_{h,n})\right)\tilde{\omega}_h^2$.

### B. Stability Analysis

*Theorem 1:* If the controller and adaptive law are formulated as (12), then it can be concluded that all signals within the MASs (1) exhibit SGUUB behavior, and the synchronization errors of all agents have the capability to converge to a narrow region around the origin.

*Proof*: Construct the Lyapunov function $V^* = \sum_{h=1}^{\mathsf{M}} V_{h,n}$, it is known that $\dot{V}_{h,n}$ can be restated as

$$\dot{V}_{h,n} \leq -\breve{\mathcal{T}}_h V_{h,n} + \breve{\mathcal{P}}_h \quad (13)$$

where $\breve{\mathcal{T}}_h = \min\{\vartheta_{h,o}, \vartheta_{h,n}, \frac{\chi_{h,o}}{2\varrho_{h,o}}, \left(\frac{\chi_{h,n}}{2\varrho_{h,n}} - \frac{|\kappa_h|}{2}\right), \left(2\kappa_h\ell_h - \ell_h^2 - |\kappa_h|\phi_{h,n}^{\mathrm{T}}(\bar{x}_{h,n})\phi_{h,n}(\bar{x}_{h,n}) - 1\right)\}$ with $o = 1,\ldots,n-1$ and $\breve{\mathcal{P}}_h = \sum_{o=1}^{n-1} \frac{\bar{\varepsilon}_{h,o}^2}{2} + \sum_{o=2}^{n-1} \frac{\bar{\mathbf{K}}_{h,f_{o-1}}}{2} + \sum_{o=1}^n \frac{\chi_{h,o}\zeta_{h,o}^2}{2\varrho_{h,o}} + \frac{1}{2}\dot{\omega}_h^2$.

After that, it can be inferred that $\dot{V}^* \leq -\breve{\mathcal{T}}V^* + \breve{\mathcal{P}}$, where $\breve{\mathcal{T}} = \min\{\breve{\mathcal{T}}_h\}$ with $h = 1,\ldots,\mathsf{M}$ and $\breve{\mathcal{P}} = \sum_{h=1}^{\mathsf{M}} \breve{\mathcal{P}}_h$. With mathematical calculations, it follows that

$$0 \leq V^* \leq V^*(0)e^{-\breve{\mathcal{T}}t} + \frac{\breve{\mathcal{P}}}{\breve{\mathcal{T}}}\left(1 - e^{-\breve{\mathcal{T}}t}\right) \quad (14)$$

Based on the above analysis and Lemma 2, it can be justified that the synchronization errors between agents are SGUUB.

## V. SIMULATION RESULTS

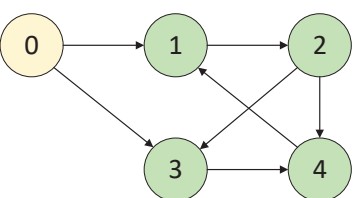

Fig. 1. Communication topology

In the given example, an MAS comprising five nodes is examined. This MAS is composed of four followers (nodes 1-4) and a leader (node 0) arranged on a directed graph, with the communication topology depicted in Fig. 1.

Consider a mass-spring-damper system [9] with dynamic $M\ddot{X} + B\dot{X} + RX = F$. Let $x_{h,1} = X$, $x_{h,2} = \dot{X}$, and $u_h = F$, then the dynamic of $h$th ($h = 1,2,3,4$) follower subject to external disturbance is modelled as

$$\begin{cases} \dot{x}_{h,1} = x_{h,2} \\ \dot{x}_{h,2} = \frac{1}{M}u_h - \frac{R}{M}x_{h,1} - \frac{B}{M}x_{h,2} + \cos(t) \\ y_h = x_{h,1} \end{cases} \quad (15)$$

The actual parameters are specified as $M = 1$ kg, $R = 3$ N/m and $B = 0.5$ Ns/m. The masked leader signal is established as $y_\pi = 2\sin(t) + 2\underline{w}(t)\exp(-0.1\underline{x}(t)t)\Pi(t)$ with $T_\epsilon = 3$. The initial values of the states within MASs are selected as $x_{1,1}(0) = 2$, $x_{2,1}(0) = 0.3$, $x_{3,1}(0) = -0.1$, $x_{4,1}(0) = -0.2$, $x_{1,2}(0) = 0.5$, $x_{2,2}(0) = 1$, $x_{3,2}(0) = 1$, $x_{4,2}(0) = 1$. For $h$th ($h = 1,2,3,4$) follower, $\hat{\theta}_{h,1}(0) = \hat{\theta}_{h,2}(0) = 0.1$. The initial values of the second-order sliding film filter are assigned as $\mathbf{K}_{h,k10}(0) = 0.1$ and $\mathbf{K}_{h,k20}(0) = 0.1$ for $h = 1,2$.

The relevant parameters of the adaptive law are given by $\varrho_{h,1} = \varrho_{h,2} = 12$ and $\chi_{i,1} = \chi_{i,2} = 2$. The controller parameters are designed as $\vartheta_{1,1} = 39.5$, $\vartheta_{2,1} = 19.5$, $\vartheta_{3,1} = 39.5$, $\vartheta_{4,1} = 39.5$, $\vartheta_{1,2} = 79.5$, $\vartheta_{2,2} = 79.5$, $\vartheta_{3,2} = 79.5$, $\vartheta_{4,2} = 79.5$. The informations of the disturbance observer are configured as $\kappa_1 = \kappa_2 = \kappa_3 = \kappa_4 = 0.01$, $\ell_1(0) = \ell_2(0) = \ell_3(0) = \ell_4(0) = 1$, $s_1^{\mathrm{L}} = s_2^{\mathrm{L}} = s_3^{\mathrm{L}} = s_4^{\mathrm{L}} = 1$.

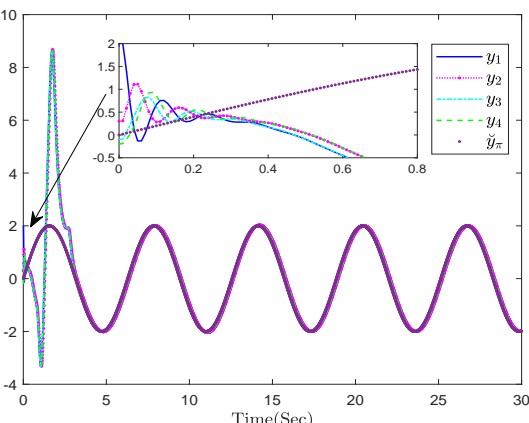

Fig. 2. The trajectories of the followers's signals and the leader's signal under privacy protection.

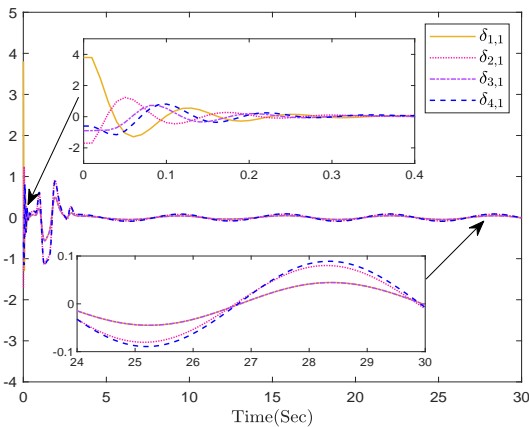

Fig. 3. The synchronization errors of the four followers.

Figs. 2 and 3 show the simulation results. There is seen in Fig. 2 that after 4 seconds the trajectories of all the agents are able to fulfil the cooperative goal of validly following the leader for which a communication link exists. Before 3 seconds it can be observed that the privacy-preserving mechanism is effective, and the followers are biased by the information they receive to the extent that the trajectories fluctuate significantly. Moreover, this fluctuation is bounded and controllable and does not lead to the collapse of the MASs. Information about the synchronisation errors is demonstrated in Fig. 3, where it can be found that when the system is relatively stable, the overall error distribution is basically in the

range of $-0.1$ to $0.1$. This illustrates that the method proposed in this paper works effectively in achieving the established control objectives.

## VI. Conclusions

In this paper, the privacy-preserving adaptive tracking control problem has been researched for MASs subject to unknown external disturbances. The future goal of our work is to apply the preassigned-time privacy-preserving mechanism to more control methods and control systems.

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
