# OpenReview forum: "Privacy-Preserving Adaptive Tracking Control for Multiagent Systems"
_IEEE.org/ICIST/2024/Conference — IEEE ICIST 2024 Conference Submission_

### Official Review · Reviewer_xRQ8 · 2024-08-20
**Lack innovation, not sufficiently important, not align**

**Rating:** 2
**Confidence:** 5

**Review:**

The findings of this work lack innovation or are not sufficiently important to the academic community. This requires reconsideration of the research question and results to ensure that a substantive contribution to the academy and the field can be made.  Meanwhile, The topic of the manuscript does not align with the scope or interests of the journal. On balance, the article will be rejected.

---

### Official Review · Reviewer_fWRV · 2024-08-21
**Lack of innovation**

**Rating:** 3
**Confidence:** 5

**Review:**

First， the main controbation of this paper is not clear. Second, the literature review in the Introduction surveys some previous work on the solutions and methods to address privacy-preserving adaptive tracking control, while comparation among these methods which is the most important part is neglected. So, this article is not recommended for acceptance.

---

### Official Review · Reviewer_jfGT · 2024-08-22
**Privacy-Preserving Adaptive Tracking Control for Multiagent Systems**

**Rating:** 6
**Confidence:** 5

**Review:**

The article presents a privacy-preserving adaptive tracking control method for nonlinear multi-agent systems (MASs) subject to unknown disturbances. The key innovation is the introduction of a masking function based on the Lorenz system, which provides unpredictability and ensures privacy protection within a user-defined timeframe. Additionally, an adaptive disturbance observer is designed using a dynamically adjustable gain parameter to handle unknown disturbances and improve system performance. The effectiveness of the proposed control scheme is validated through simulations, demonstrating its ability to maintain system stability while preserving privacy. On the whole, the paper is good, but there are still some problems that need to be corrected.

Comments:

1.The paper introduces a novel privacy-preserving mechanism and an adaptive disturbance observer. However, the distinction between these contributions and existing methods could be more explicitly articulated to highlight the innovation.

2.While the simulation results effectively demonstrate the proposed method's performance, a more detailed comparison with other existing methods would strengthen the paper's argument for the proposed approach's superiority.

3.Have you considered using other diagrams to illustrate your method? I think that adding a block diagram could help in understanding the principles of the method used.

4.The research content is well constructed. But there are grammatical errors and incorrect English writing in the current manuscript. Please make the necessary corrections.

5.The formula symbol is more complex, it is recommended to carefully check and modify the full text.

---

### Decision · Program_Chairs · 2024-09-08

Reject